# Cellular and Molecular Mechanisms Associating Obesity to Bone Loss

**DOI:** 10.3390/cells12040521

**Published:** 2023-02-05

**Authors:** Yasmin Silva Forte, Mariana Renovato-Martins, Christina Barja-Fidalgo

**Affiliations:** 1Laboratory of Cellular & Molecular Pharmacology, IBRAG, Universidade do Estado do Rio de Janeiro, Rio de Janeiro 20551-030, Brazil; 2Laboratory of Inflammation & Metabolism, Instituto de Biologia, Universidade Federal Fluminense, Niterói 24210-201, Brazil

**Keywords:** obesity, inflammation, adipose tissue, extracellular vesicles, adipokines, osteoblast, osteoclast, osteocyte, Nc-RNAS, gut microbiota

## Abstract

Obesity is an alarming disease that favors the upset of other illnesses and enhances mortality. It is spreading fast worldwide may affect more than 1 billion people by 2030. The imbalance between excessive food ingestion and less energy expenditure leads to pathological adipose tissue expansion, characterized by increased production of proinflammatory mediators with harmful interferences in the whole organism. Bone tissue is one of those target tissues in obesity. Bone is a mineralized connective tissue that is constantly renewed to maintain its mechanical properties. Osteoblasts are responsible for extracellular matrix synthesis, while osteoclasts resorb damaged bone, and the osteocytes have a regulatory role in this process, releasing growth factors and other proteins. A balanced activity among these actors is necessary for healthy bone remodeling. In obesity, several mechanisms may trigger incorrect remodeling, increasing bone resorption to the detriment of bone formation rates. Thus, excessive weight gain may represent higher bone fragility and fracture risk. This review highlights recent insights on the central mechanisms related to obesity-associated abnormal bone. Publications from the last ten years have shown that the main molecular mechanisms associated with obesity and bone loss involve: proinflammatory adipokines and osteokines production, oxidative stress, non-coding RNA interference, insulin resistance, and changes in gut microbiota. The data collection unveils new targets for prevention and putative therapeutic tools against unbalancing bone metabolism during obesity.

## 1. Introduction

Obesity is a disease characterized by abnormal or excessive fat accumulation that can harm health [1]. By 2030, it is estimated that over 1 billion people globally will become obese, with 20% in the female population and 15% in the male population [2]. Changes in society eating habits, following Western diet patterns, and reduced physical activity support its worldwide spread [3]. Obesity is associated with many diseases, such as diabetes, osteoarthritis, cardiovascular disorders, and some types of cancers, contributing to a further rise in the mortality rate [4,5,6]. It is estimated that 2.8 million people die annually from being overweight and obese [7].

The onset and development of the obesity condition depends on microenvironmental, metabolic, genetic, and behavioral factors [3]. The imbalance between increased food intake and decreased energy expenditure leads to alterations in adipose tissue distribution and secretion [8,9]. The hypertrophy of adipocytes, caused by excessive fat accumulation, enhances the expression of proinflammatory mediators, such as TNF-α, IL1β, IL6, resistin, and leptin in the adipose tissue. Infiltration of M1-like macrophages and hypoxic conditions due to insufficient blood vessel availability favors adipose tissue inflammation [10,11,12]. The release of adipose tissue-derived mediators in the bloodstream promotes a systemic low-grade inflammatory state that may reach other sites, such as the hypothalamus, skeletal muscle, pancreas, liver, and bone [13,14,15,16,17,18].

The obese adipose tissue also secretes increasing levels of extracellular vesicles (EVs) containing a variety of molecular cargos that may contribute to systemic alterations [19]. These EVs provide cellular communication, acting paracrine and endocrine, through the delivery of different proteins, transmembrane receptors, lipids, and nucleic acids [20,21]. We have recently demonstrated that the obese human adipose tissue releases more significant amounts of EVs than eutrophic adipose tissue [22]. Those EVs may regulate physiological and pathological processes, such as bone remodeling, cancer progression, embryonic development, tissue repair, and other biological phenomena [21,23,24]. The obese adipose tissue vesicles elicit a proinflammatory phenotype in monocytes and increase breast cancer malignancy [22,25]. However, little is known about obese adipose tissue EV content and their participation in bone remodeling. During obesity, non-coding RNAs (Nc-RNA) are released within extracellular vesicles or freely and contribute to systemic alterations. Nc-RNAs have been suggested as putative tools for diagnostics and therapeutics [26,27]. Nevertheless, the role of Nc-RNAs content from obese adipose tissue VEs on bone remodeling is still poorly investigated.

Bone is a dynamic calcified tissue involved in physical support, visceral protection, mineral storage, hematopoiesis, and endocrine activity [28,29,30]. Bone tissue renewal is a very balanced process where hormones, cytokines, and growth factors regulate bone formation and bone resorption [31,32]. Osteoblasts are bone-forming cells derived from mesenchymal stem cells (MSCs). They produce organic extracellular matrix elements and promote their further mineralization [33,34]. On counterbalance, osteoclasts are multinucleated cells derived from the monocyte/macrophage lineage, able to resorb bone by enzymatic digestion. Bone resorption occurs predominantly by cathepsin-K and MMP-9 actions in an acidic microenvironment [35,36]. A fine balance between osteoclast and osteoblast activities controls healthy bone remodeling [37]. Additionally, osteocytes are bone cells sensitive to mechanical loads that orchestrate regulatory mechanisms to enable bone turnover. The process involves the activation of signaling routes, such as WNT, and the release of mediators, such as bone morphogenetic proteins (BMP), nitric oxide (NO), prostaglandin E2 (PGE2), sclerostatin, FGF-23 and RANKL [38,39].

Due to the systemic inflammatory condition, obesity can disrupt the balance of bone remodeling, dysregulating bone homeostasis and allowing bone loss. Two main pathways drive this condition: (i) the release of proinflammatory mediators from obese adipose tissue that reduces osteoblast activity while increasing osteoclast differentiation and resorption; and (ii) the differentiation of MSCs towards osteogenic lineage is reduced during obesity, while adipogenic differentiation is enhanced [40,41,42]. Nevertheless, other systemic alterations may also have relevance in this process, including oxidative stress, gut and microbiota physiology, vesiculated or free Nc-RNA, and hormonal changes. This review gathered the main mechanisms involved in the crosstalk between obese adipose tissue and bones that lead to bone loss.

## 2. Materials and Methods

The following keywords, in combination or isolated forms, were searched on PubMed and Google Scholar platforms: “Obesity”; “TNF-α”; “IL-1β”, “IL-6”; “Leptin”; “Adiponectin”; “Lipocalin 2”; “gut microbiota”; “Lipopolysaccharide (LPS)”; “Reactive Oxygen Species (ROS)”; “osteocalcin”; “glucose metabolism”; “osteoblast”; “osteoclast”; “bone loss”; “NcRNA” and “MiRNA.” Reviews, repeated studies, and articles with divergent subjects were excluded. Experimental and cohort analyses published recently and suitable for our research were included. The figures were created using LibreOffice IMPRESS 4.7 program, and a few images from the copyright-free image websites PIXABAY and FLATICON were used for composition.

## 3. Results

### 3.1. Proinflammatory Cytokines in Bone Metabolism

Adipose tissue expansion increases pro-inflammatory cytokine expression and secretion by hypertrophic adipocytes and infiltration of M1-like macrophages and other immune cells (Figure 1). Cytokines, such as TNF-α, IL-1β, and IL-6 released by obese adipose tissue can interfere with bone cell homeostasis. Acting in isolation or in concert, they trigger intracellular signaling pathways that may lead to bone loss.

TNF-α is a well know enhancer of osteoclastogenesis [43]. In vitro experiments showed that stimulation of osteoclast precursors with TNF-α and RANKL increased the number of multinucleated TRAP-positive cells [44,45]. Additionally, in vivo treatment with TNF-α increased the osteoclast numbers in mice calvaria [46]. In this context, Shinohara and collaborators (2015) demonstrated that TNF-α-induced osteoclast formation depends on double-stranded RNA-dependent protein kinase [44]. Other studies showed that TNF-α increased c-kit+/c-fms+ osteoclast precursors in high-fat diet (HFD) fed obese mice and augmented RANK expression in bone marrow macrophages [47,48]. Increased RANKL expression in TNF-α stimulated osteocytes contributed to osteoclast differentiation in vitro and in vivo [43].

In parallel, TNF-α can also modify osteoblast activity: In MC3T3-E1, osteoblasts or MSCs induced osteogenic differentiation, TNF-α decreased the expression of osteogenic transcription factors RUNX2 and Osterix, reducing the mineralization and expression of bone markers, osteocalcin (OCN), alkaline phosphatase (ALP), and bone sialoprotein (BSP) [49,50,51,52,53]. These effects are mediated by the inflammatory transcription factors ATF3 [51] and Nf-κB [53]. TNF-α upregulated miR-150-3p in osteoblasts, decreasing β-catenin expression, a key transcription factor for osteogenesis [52]. This cytokine can affect osteoblast viability, promoting apoptosis [54] and increasing MMP-9 expression [55]. TNF-β, another TNF family member, also showed deleterious impacts on bones, reducing RUNX2, Type I collagen (Col1A), OCN, and integrin β1 expression and decreasing mineralization in MSCs cultured in the osteogenic medium [56].

In vivo studies have demonstrated TNF-α-related bone loss in obese mice [57,58]. A palmitic acid-enriched HFD enhanced TNF-α serum levels in obese mice, parallel with a decrease in bone turnover markers, P1NP e CTX1 [57]. Furthermore, mice fed with a HFD presented lower trabeculae numbers and thickness and reduced trabecular bone volume, compared to lean controls. Obese TNF-α knockout mice showed less bone loss, a reduced number of femoral osteoclasts, and enhanced RUNX2 expression in MSCs, compared to HFD wildtype [58].

Increased circulating levels of IL-1β are a hallmark of the chronic, low-grade inflammation associated with obesity and related diseases [59]. IL-1β is a known inducer of osteoclast migration and resorption [60,61,62]. Combined with RANKL, IL-1β significantly increased osteoclast formation in vitro, enhancing TRAP staining and the expression of osteoclast markers, cathepsin K, OSCAR, NFATC1, Cfos, and DC-STAMP [61,62,63,64,65]. Though IL-1β increases osteoclast formation, its effect depends on the precursor subsets. Comparing the early myeloid blasts and monocytes, IL-1β increased osteoclast formation and cell diameter, mostly in myeloid blasts [61]. IL-1β can also harm other bone cells, inducing apoptosis in the MLO-Y4 osteocytic cell line [66] and MC3T3-E1 osteoblasts [67]. Osteoblasts stimulated with IL-1β exhibited a lower migration capacity [68] and released more IL-6 [69]. Indirectly, IL-1β promoted an osteoclastic supportive phenotype in osteocytes, enhancing RANKL expression [67]. Apart from triggering specific mechanisms, IL-1β also shares some signaling mechanisms with TNF-α. Lee and collaborators (2017) have shown that treatment with IL-1β and TNF-α raised CCR7 expression in both precursors and differentiated osteoclasts, increasing their migratory activity toward CCL19 and CCL21 chemokines upregulated in rheumatoid arthritis [60]. Adseverin, an actin-binding protein that regulates cell differentiation and fusion, modulates IL-1β- and TNF-α-induced osteoclastogenesis [63]. Nevertheless, Hah and collaborators (2013) showed that both IL-1β and TNF-α increased ALP activity and mineralization of periosteal osteoblasts without modifying RUNX2 and OCN expression [70].

Although osteoblasts are characterized by their activity of extracellular matrix synthesis, IL-1β enhanced metalloprotease production by osteoblasts. Ozeki and collaborators (2014) demonstrated that this cytokine potentiated MMP13 expression through ADAM28 upregulation in osteoblast-like cells [67]. Yang and collaborators (2011) have shown that IL-1β amplified the expression of MMP-9 and MMP-13 in osteoblasts [71], further contributing to bone destruction in inflammatory diseases. IL-1β may also interfere with hormonal factors produced by bone tissue, such as FGF23, which orchestrates vitamin D and phosphate serum levels [72,73]. An imbalance in FGF23 regulation, mainly due to its exceeding action, leads to a pathological mineralization process that weakens bones [74]. Corroborating in vivo studies that described greater serum levels of FGF23 in IL-1β treated mice [73], the treatment of bone slices in vitro increased hormone secretion [72]. Additionally, He and collaborators (2020) demonstrated an association between IL-1β genetic variants and osteoporosis predisposition [75]. IL-1β variants rs1143627, rs16944, and rs1143623 are related to elevated susceptibility to osteoporosis, especially in women older than 60 or with a BMI greater than 24 kg/m^2^ [75]. In obese mice fed a 10% corn oil-based diet, Halade and colleagues (2011) observed that bone marrow adipose tissue secreted higher concentrations of IL-1β, TNF-A, and IL-6, which increased the expression of RANK-L in stromal cells favoring osteoclast formation [76].

IL-6 is a cytokine with pleiotropic actions secreted by several cell types, including adipocytes, and its plasma levels are significantly upregulated during obesity [77]. In bone tissue, such as IL-1β and TNF-α, IL-6 indirectly stimulated osteoclastogenesis, increasing RANKL expression in osteocytes and osteoblasts [78,79,80]. Using an in vitro model of bone loss, neutralizing IL-6 antibodies had protective effects against osteoporosis, enhancing bone mineral density, trabecular number, and thickness [81]. IL-6 inhibition decreased osteoblasts’ RANKL/OPG ratio and osteoclast differentiation in a microgravity model [81]. IL-6 also inhibited osteoblast activity by downregulating the expression of the osteogenic transcription factor RUNX2 [79,81,82]. IL-6 knockout osteoblasts presented higher ALP activity and RUNX2 expression than wild-type ones. Moreover, obese IL-6 knockout mice exhibited increased trabecular bone volume, number, and thickness than HFD wild-type mice [82].

Few publications have described a combined effect of proinflammatory cytokines on bone cells. Stimulation with IL-1β, TNF-α, and IL-6 together in bone chips enhanced IL- 1β, IL-6, IL-8, TNF- α, FGF23, SOST, and OPG expression by bone cells [83]. Yokota and collaborators (2013) showed that TNF-α and IL-6 supracalvarial injections in mice raised osteoclast formation, compared to isolated injections of both cytokines [84]. Together, these data suggest the pro-osteoclastic and anti-osteoblastic effects of proinflammatory cytokines and their importance in obesity-associated bone loss (Figure 2).

### 3.2. Adipokines in Bone Remodeling

Leptin and adiponectin are the two main characteristic adipokines produced by adipose tissue [85]. Leptin is also synthesized by the gastrointestinal tract, muscle, and brain [86] and regulates food intake, body weight, reproductive system, proinflammatory responses, and lipid metabolism [87]. Adiponectin is mostly synthesized by the adipocytes but also by myocytes and endothelial cells, acting to increase insulin sensibility, fatty acid oxidation, and anti-inflammatory, anti-diabetic, and anti-atherogenic effects [88,89]. Evidence has demonstrated that leptin and adiponectin have regulatory effects on bone metabolism [90,91].

Increased adiponectin and lower leptin secretion characterize homeostasis of healthy lean adipose tissue. In contrast, in obesity, the pathological adipose tissue expansion, with hypertrophy of adipocytes, unbalance that pattern, and the obese adipose tissue releases more leptin and less adiponectin [85,92]. These alterations contribute to systemic inflammation, insulin resistance, and skeletal disturbances during obesity [93,94].

The relevance of adiponectin signaling on bone tissue homeostasis has been evidenced in mice knockout models and cell culture systems. The deficiency of adiponectin in vivo resulted in bone loss, characterized by decreased bone volume/tissue volume ratio in the number of trabeculae, and the increase in trabecular separation [95], without alteration in bone mineral density [96]. Lack of adiponectin also compromised osteointegration of bone explants in deficient mice [97]. Abbott and collaborators (2015) described a reduction of 15% in the tibial fractional bone volume of mice with suppression of adiponectin in adipocytes, compared to controls, probably due to a higher differentiation or proliferation of bone marrow adipocytes in those animals [98]. Adiponectin seems to have a determinative impact on osteoclast formation and function. A decrease in adiponectin availability enhanced the number of osteoclasts in the femur of knockout mice [96,97]. The treatment of osteoclast precursors with recombinant adiponectin reduced multinucleated TRAP+ cell formation and expression of osteoclast markers, such as cathepsin K, OSCAR, and NFAT2 [97,98,99,100,101,102,103]. It Interfered in osteoclastogenesis signaling pathways, such as NF-κB, MAPK, and AKT [8,9]. Adiponectin reduced the viability of osteoclast progenitors, as shown by apoptosis and proliferation assays [97,101].

Unlike osteoclasts, adiponectin improved osteoblasts’ functions. Adiponectin treatment increased Osterix and BSP promoter activities in MC3T3-E1 murine osteoblasts [97] and enhanced extracellular matrix mineralization of primary osteoblasts and pre-osteoblasts [102,104]. Pacheco-Pantoja and collaborators (2014) demonstrated a reduced ALP activity in SAOS-2 osteoblasts [105]. Much evidence has demonstrated the role of adiponectin in osteoblast/adipocyte differentiation balance. In a model of ovariectomy-induced bone loss, the delivery of adiponectin by an adenovirus system raised mRNA levels of OCN, RUNX2, and ALP and reduced PPARγ and CEBP on bone tissue [106]. Adiporon, an agonist of adiponectin, favored osteogenic differentiation of bone marrow- and adipose tissue-derived MSCs, in vitro and in vivo, while decreasing adipogenic differentiation [102].

Adiponectin gene expression positively correlated with RUNX2 and negatively with RANKL/OPG ratio in the bones of healthy and osteoporotic patients [107]. In multiple myeloma patients, adiponectin was positively correlated with OCN, a bone formation marker, and negatively with the resorption marker CTX-1 [99]. Hence, adiponectin deficiency may be an essential mechanism of bone remodeling disbalance, including resorption stimulation and inhibition of bone formation in subjects with obesity.

In contrast with the consolidated data obtained with adiponectin studies, the effects of leptin on bone tissue are still conflicting. The contrasts occurred mainly between experimental and human models. Yue and collaborators (2016) showed that obese leptin receptor knockout mice presented an increased trabecular number and thickness, mineral apposition and formation rates, and decreased bone marrow adipocyte number, compared to the wild-type obese group [108]. Supporting the data showing a link between leptin and bone resorption, rats submitted for eight weeks to a HFD exhibited a lower trabecular number/thickness and bone mineral density (BMD), in parallel with serum leptin levels positively correlating with RANK levels [109]. In an innovative model, Carnovali and collaborators (2017) showed that adult male zebrafish fed a HFD presented increased ratio leptin/adiponectin serum levels, accompanied by a 12.6% reduction of the bone mineralized area, decrease in ALP activity, and increased TRAP activity and analog RANKL expression, compared to animals fed a standard diet [110]. In contrast, Feresin et al. (2014) did not find differences in bone parameters between Zucker rats fed a normal or HFD [111]. Philbrick and collaborators (2017) showed that administration of leptin in ob/ob mice increased bone formation rates and CTX-1 serum levels [112] and accelerated bone repair after the surgical fracture model [113].

Discordant information is also found in studies on the role of leptin in MSCs. While Tencerova and collaborators (2019) found that LepR+ bone marrow-MSCs from patients with obesity are associated with accelerated senescence and bone fragility [114], Li’s group showed that human MSCs differentiated toward osteogenic lineage in the presence of leptin had an increased ALP activity, compared with untreated cells [115]. Furthermore, analysis in a cohort of adolescents has demonstrated that bone mineral density (BMD) was positively correlated with serum leptin and fat mass [116]. Similarly, lumbar and hip BMD of Japanese postmenopausal women presented a positive correlation with leptin and a negative one with adiponectin [117]. Moreover, studies comparing children with normal weight and with obesity concluded that leptin levels are inversely related to tibial trabecular thickness [118].

Conflicting data may be explained by the distinct signaling pathways activated by leptin interfering differently in trabecular and cortical bone formation. Leptin regulates bone metabolism through direct anabolic signaling on bone cells and indirectly by the central nervous system. In the hypothalamus, leptin-mediated signaling regulates downstream effectors, such as neuropeptide-Y. In the presence of leptin, neuropeptide Y activity is inhibited, which causes an increase in cortical bone mass [119]. Wong and collaborators (2013) showed that knockout mice for leptin and neuropeptide Y have more significant gains in the cortical bone mass of long bones and vertebrae and a better mineral apposition rate than ob/ob mice [120].

Interestingly, in obese neuropeptide-Y-knockout mice, the cortical bone loss and fat accrual were suppressed, compared to wild types [121]. This information is relevant because, in subjects with obesity with leptin receptor resistance, the lack of signaling in bone cells and the reduction or inhibition of neuropeptide Y can contribute to deleterious effects on bones. However, the exact consequence of leptin signaling on bone tissue in obesity is still unclear.

In addition to leptin and adiponectin, ghrelin and glucagon-like peptide 1 (GLP-1), are altered during obesity and also influence bone metabolism [122], and may have a role in bone remodeling. Ghrelin is produced by the stomach, acting on appetite regulation and energy metabolism, improves the cardiovascular function, and can act in bone formation [123]. Erener et al. (2021) showed that ghrelin administration in mice enhanced the healing process of bone fractures [123]. Notably, in patients with osteopenia, the systemic ghrelin concentration are inversely related to BMI values, and the osteoblasts isolated from those patients were resistant to ghrelin-mediated signaling [124].

The therapeutic use of GLP-1 receptor agonists has been well described to improve for the metabolic parameters of patients with obesity and insulin resistance [125]. Additionally, activation of GLP-1 receptor on MSCs induced osteogenic differentiation via PKA/β-catenin signaling activation [126], and improved bone composition in obese animals [127]. Together, the data point to ghrelin and GLP-1 as worthy candidates for pharmacological intervention for controlling obesity-associated bone loss.

### 3.3. Muscle-Fat-Bone Crosstalk during Obesity

Bones and skeletal muscles are associated tissues that promote locomotion, and the mechanical forces driven by muscle contraction are vital factors that allow bone formation. However, the muscle-bone crosstalk is not limited to structural functions. Myokines produced and released in response to muscular contractions have osteoanabolic effects and are essential for maintaining bone metabolism [128]. Chronic low-grade inflammation in patients with obesity modifies the crosstalk between muscle-fat-bone [129]. Fatty acids released by intramuscular adipose tissue induced lipotoxic effects on osteoblasts, osteocytes, and myocytes [130]. Disturbances in muscle metabolism caused by overweight/obesity may impact bone tissue, contributing to osteosarcopenic obesity syndrome.

The sedentary profile of people with obesity reduces the anti-inflammatory activity performed by myokines, such as irisin. This exercise-triggered myokine promotes white adipose tissue browning, energy expenditure and stimulates bone formation [131]. Knockout mice for the irisin precursor FNDC5 (fibronectin-type III domain-containing 5) showed bone loss mediated by increased resorption activity and reduced bone formation [132,133]. Irisin stimulates osteoblast differentiation through αV integrin activating-BMP/SMAD pathway [132], leading to activation of RUNX1/2 via focal adhesion kinase [133]. The administration of irisin in high-fat diet-induced obese mice increased insulin, ghrelin, and BMP4 circulating concentrations [134]. Obese rodents subjected to swimming showed improvement in bone loss, simultaneously with increased systemic irisin production and release [135]. Addressing the crosstalk among bone, muscle, and adipose tissue may be a successful pharmacological strategy for treating bone and muscle loss in obesity.

### 3.4. Gut and Microbiota Alterations Influencing Bone Metabolism

The crosstalk between obesity and intestinal disorders still raises doubts about the cause and effect mechanisms since weight gain and intestinal damage are complementary events. Studies using germ-free mice proved the importance of gut microbiota regulation on fat accrual and obesity development, where a lack of microorganisms reduced 42% of total body fat [136]. Additionally, microbiota transplantation from obese to germ-free mice increased body fat, compared to lean microbiota transplantation [137]. Concurrently, it is well known that environmental factors, such as the Western diet compromise gut microbiota quality, favoring weight gain [138]. In individuals with obesity, proinflammatory cytokines, such as TNF-α and IFN-γ produced by adipose tissue interfere with tight junctions of the intestinal barrier, promoting increased permeability [139]. As a result, components of gram-negative bacteria cell wall, such as the endotoxin lipopolysaccharide (LPS), can bypass the loosened intestinal cell junctions and enters the bloodstream. Enhanced LPS serum levels contribute to the exacerbation of inflammation, insulin resistance, adipose tissue pathological expansion, and other systemic damage [140]. Therefore, obesity interferes with bone homeostasis through gut modifications in two principal ways: The first is the change in the intestinal microbiota profile, and the second is the elevated serum endotoxin levels (Figure 3).

Luo and collaborators (2015) demonstrated that fecal microbiota transplantation (FMT) from HFD mice to normal diet mice decreased the osteoblast surface and RUNX2 gene expression [141]. In a periodontitis model, FMT from obese mice also increased the periodontal tissue inflammation and bone destruction [142].

Among several types of bacteria in the gut microbiota, two main phyla stand out, Bacteroidetes and Firmicutes. In healthy individuals, there is usually a favoring in Bacteroidetes growth. However, subjects with obesity tend to have a more significant presence of Firmicutes [143]. Changes in Firmicutes/Bacteroidetes ratio have been associated with inflammatory conditions and systemic effects [144]. Mice fed a high-fat diet presented an enhancement of the Firmicutes/Bacteroidetes ratio, which correlated negatively with bone volume [145]. In a complementary way, another study comparing sham and ovariectomized mice showed that Bacteroidetes positively correlate with bone volume/tissue volume, bone mineral density, and trabecular number. At the same time, Firmicutes are negatively related to the same bone microstructure parameters [146]. Another phylum, Proteobacteria, also appears to be associated with obesity-bone loss. In a Mexican population study, children with obesity presented an abundance of Proteobacteria phylum, compared to normal-weight children in stool samples [147]. At the same time, more significant amounts of Proteobacteria were found in fecal samples of low BMD women, compared to average BMD women [148].

Along with Bacteroidetes, obesity reduces other beneficial bacteria genera, such as Akkermansia, Bifidobacterium, and Lactobacillus [149], compromising healthy bone remodeling. The beneficial bone effects of Akkermansia were shown to be mediated by EVs release, which enhanced osteoblast differentiation and mineralization while decreasing osteoclasts formation [150]. Akkermansia muciniphila administration reduced local inflammation and bone destruction in a model of infection-induced bone loss [151]. Moreover, patients with osteopenia had a lower frequency of Akkermansia bacteria in fecal samples than healthy patients, although the comparison did not establish a statistical difference [152].

Bifidobacterium pseudocatenulatum administration in HFD-induced obese mice improved the trabecular parameters, compared to non-treated obese mice. It increased the genic expression of Wnt1, a ligand that induces osteogenesis through the β-catenin signaling pathway. Serum dosages also showed that Bifidobacterium enhanced the bone formation marker OCN and decreased the bone resorption marker CTX-1 in HFD mice [153]. Corroborating this data, administration in obese mice of VSL#3, a commercial probiotic formulated predominantly by Bifidobacterium and Lactobacillus species, increased the trabecular bone volume and bone mechanical strength, compared to the obese vehicle group. Bone formation was due to the inhibitory effect upon osteoclast differentiation and the favoring of osteoblast activity through histone methylation [154]. The treatment of obese insulin-resistant mice with Lactobacillus paracasei, alone or conjugated with the prebiotic Xylooligosacharide (XOS), reduced systemic inflammation [155,156] and the eroded bone surfaces, reestablishing bone mineral apposition and formation, harmed by obesity [156]. It was reported that the probiotic Lactobacillus rhamnosus GG increases bone mass by modifying immune responses by enhancing serum butyrate concentrations. Butyrate raises the frequency of T-regulatory lymphocytes in the gut and bone marrow which, in return, stimulates TCD8+ Lymphocytes to release wnt10b, a positive regulator of bone formation [157]. Obesity promotes the reduction of butyrate-producing bacteria in the gut [158], which may be an additional interference with bone metabolism.

Due to obesity-induced dysbiosis, prebiotics have become an alternative therapy to revert the prejudicial impacts of imbalanced microbiota on different systems. In addition to XOS [155,156], other prebiotics, such as Fructooligosaccharide and Galactooligosaccharide reduced bone loss in HFD mice, increasing osteoblast differentiation and decreasing osteoclast and adipocyte formation. The prebiotics also ameliorates gut dysbiosis by reducing the Firmicutes/Bacteroidetes ratio, normalizing intestinal permeability [159]. Combined with exercise, fiber prebiotics prevented knee joint damage, insulin resistance, and endotoxemia in high-fat and high-sucrose diet (HFS) mice [160].

Changes in gut permeability observed during obesity amplify the availability of LPS—a potent inducer of bone degradation [155,160]. In animal models of bone loss, systemic (intraperitoneal) or local (in calvaria) administration of LPS promoted reduction in BMD, trabecular thickness, and bone volume while increasing the number of TRAP+ cells on the bone surface and CTX-1 serum concentration [161,162,163,164]. In female Zucker fat rats, LPS administration also decreased bone volume [164]. LPS stimulation of MC3T3-E1 osteoblasts reduced cell viability with an increased genic expression of caspase 3 and Bax [165], decreasing the ALP activity and expression of osteogenic markers [165,166,167]. Moreover, LPS treatment modifies osteoblasts to better support osteoclastogenesis. Primary osteoblasts from LPS-treated mice had enhanced expression of RANKL, compared to mice that received saline injections [163]. Moreover, LPS increased EphA2 expression in both osteoblasts and osteoclasts in vitro [166]. Signaling mediated by EphA2 and its ligand EphrinA2 promoted osteoclast formation and inhibited osteoblastogenesis [168]. Together with RANKL, LPS potentiated multinucleated TRAP+ cell differentiation in vitro and raised the expression of cathepsin K, MMP-9, and NFATc1 [162,166,167]. Xing and collaborators (2011) also showed that LPS promotes pre-osteoclasts migration and differentiation through CXCR4 expression via TLR4 [169]. This whole dataset suggests therapeutic strategies that target the resolution of obesity-induced gut dysbiosis and compromised intestinal barrier integrity as relevant ways to combat bone loss.

### 3.5. Osteocalcin, FGF23, and Lipocalin-2 Regulations upon Glucose Metabolism

Beyond their mechanical properties associated with physical support and visceral protection, the bone tissue has other functional activities, including mineral storage, hematopoiesis support, and hormone release [28]. The endocrine function of bones was recently discovered, and the three main osteokines produced are FGF23, osteocalcin (OCN), and lipocalin-2 [170]. FGF23 is produced by osteoblasts and osteocytes and regulates phosphate resorption by renal proximal tubules and calcitriol production [171,172]. OCN, a bone formation marker, stimulates insulin sensitivity and production and increases energy expenditure [173].

Similarly, lipocalin-2 regulates insulin signaling, improves glucose tolerance, and inhibits appetite [174]. Therefore, bone tissue plays a crucial role in maintaining a healthy energy metabolism. Disturbances in bone metabolism promoted by obesity interfere with glucose metabolism and weight gain, worsening obesity conditions and aggravating bone loss (Figure 4).

OCN is produced by mature osteoblasts and assumes different activities depending on post-translational modifications. γ-carboxylated osteocalcin (GlaOC), due to carboxylation, binds to hydroxyapatite and is part of the bone extracellular matrix. Uncarboxilated osteocalcin (UnOC), released after osteoclast’s bone resorption, works as a hormone and promotes glucose tolerance [173]. Studies comparing individuals with normal weight and with obesity showed that obesity reduces UnOC serum concentration, as seen in works that compared premenopausal women, children, and men with obesity with their respective lean controls [175,176,177,178]. Additionally, Vigevano and collaborators (2021) demonstrated that, compared to healthy men with obesity, type 2 diabetes-bearing men with obesity had a lower UnOC serum concentration, increased trabecular separation at the tibia and radius, and decreased tibial failure load and stiffness [179]. In HFD-induced obese mice, administration of UnOC reduces weight and fat pad gain [180,181], insulin resistance, and the expression of inflammatory genes [181]. Moreover, it improves glucose tolerance [180,181] and increases glucose transporter type 4 (Glut4) expression in white adipose tissue of treated animals [181]. UnOC injections also prevented obesity and type 2 diabetes in HFD mice and increased the number of mitochondria in skeletal muscle [182].

Unlike OCN, FGF23 serum concentrations have been positively correlated with the BMI [183,184]. In this context, studies including pre- and post-menopause women and adult men showed that circulating FGF23 was significantly higher in individuals with obesity, compared to lean individuals [185,186]. High FGF23 availability is associated with declined metabolic function and cardiovascular disorders [183,185]. Additionally, changes in this osteokine compromise bone metabolism. Rupp and colleagues (2019) demonstrated that high serum levels of FGF23 correlated with trabecular bone loss in osteoporotic patients [74]. Moreover, in an osteoporosis mouse model, FGF23 contributed to bone loss by activating the JAK/STAT pathway [187]. Although FGF23 is widely associated with phosphate and vitamin D metabolisms, some studies have implicated this hormone in insulin resistance [183,188].

Indeed, diabetes is an additional factor that promotes osteoporosis [189] and, combined with obesity, amplifies bone loss. Deleterious effects on the bones of diabetic subjects seem to occur mainly due to the inhibition of osteoblast activity. Under high glucose conditions, in vitro, osteoblasts present reduced expression of ALP, OCN, and RUNX2, while proinflammatory cytokines IL-1β and IL-6 were increased, which is an indirect way to stimulate osteoclasts formation [190]. In diabetic mice and in vitro models, high glucose concentrations decreased osteoblast proliferation and differentiation through pyroptosis [191]. In HFD-induced obese mice, metformin administration reversed the reduction of osteoblasts in long bones [192]. Subjects with obesity may present high serum concentrations of advanced glycation end-products (AGEs) in urine samples. Osteoblasts treated with AGEs presented lower OCN and Col1A expression [178]. Despite its effect on osteoblasts, the mechanisms explaining how glucose impacts osteoclasts are still not understood. Contradictory data shows both increase [193] and inhibition [194] of osteoclast differentiation and resorption activity face high glucose concentrations in vitro.

Obesity also influences serum lipocalin-2. Despite its action in improving insulin sensitivity and appetite inhibition [174,195], patients with obesity present greater serum levels of this hormone than lean individuals [196,197,198]. This intriguing data is justified as a compensatory mechanism trying to counter the metabolic disturbance promoted by obesity [195]. Hence, increased lipocalin-2 production by bones could be a protective response to diabetes-associated bone loss. This hypothesis is supported by studies revealing that lipocalin-2 negatively regulates osteoclast formation [199]. However, many studies describe the harmful effects of excessive lipocalin-2 on bone remodeling. Some works emphasize the catabolic and inflammatory responses triggered by lipocalin-2, which impact systemically [200]. In this way, an increase in lipocalin-2 was associated with an 80 to 81% increased risk of osteopenia and hip fracture [198].

Additionally, overexpression of lipocalin-2 in bones reduced the cortical bone layer and trabecular number and increased RANKL and IL-6 expression. An increased number of osteoclasts was also identified in vivo and in vitro upon lipocalin-2 stimulation, compared to controls [201]. Osteoblasts transfected with lipocalin-2 presented lower expression of RUNX2, Osterix, and ALP [202]. Therefore, it is necessary to elucidate the role of lipocalin-2 in obesity and its systemic effects on bones to establish robust conclusions. Moreover, insulin resistance, mediated partly by UnOC and FGF23 (bone hormones) alterations in subjects with obesity, contributes to the osteoporosis scenario in obesity.

### 3.6. Oxidative Stress and Bone Remodeling

Oxidative stress is an imbalance between oxidizing agents, reactive radical or non-radical species, and antioxidant agents. In pathological situations, such as obesity, and cardiovascular and neurological diseases, reactive species production is predominant while antioxidant defense mechanisms are insufficient or reduced. Therefore, the excess of unstable molecules causes structural damage to proteins, DNA, and lipids, compromising cell integrity and functionality and leading to cell death [203,204]. Obesity induces oxidative stress in different ways, including hyperglycemia, increased plasma free fatty acids (FFAs) and endothelial dysfunction, deficiencies in minerals and vitamins, and diet composition [205]. Mitochondrial dysfunction occurs in adipocytes and immune cells within obese adipose tissue and promotes increased production of reactive oxygen species (ROS) and lipid peroxidation products [204,206]. Oxidative stress is observed in rodent models of obesity, with increased concentrations of plasmatic ROS and thiobarbituric acid reactive substances (TBARS) [207]. In subjects with obesity, subcutaneous adipose tissue samples present higher ROS levels and increased mitochondrial damage than lean ones [208].

Indeed, the release of oxidative stress mediators promotes considerable damage to bone tissue, enabling the development of osteoporosis [209]. In MSCs, enhanced levels of intracellular ROS released by NOX4 favor adipogenesis [210]. Concurrently, in a coculture system of 3T3-L1 differentiated adipocytes and MC3T3-E1 osteoblasts, adipocytes release FFAs, which promote oxidative stress in osteoblasts. In this context, osteoblasts present reduced mineralization activity, lower RUNX2, Col1A, and OCN expression, and increased apoptosis [211]. These data demonstrate that redox imbalance favors the expansion of adipose tissue while impacting bone formation.

NOX2 (GP91^PHOX^) is the catalytic subunit of NADPH oxidase that is highly expressed in immune cells. In HFD-induced obese mice, increased ROS production is observed in the bone marrow and white adipose tissue. The majority of ROS is produced by NOX2, as demonstrated in NOX2-deficient obese mice [212]. In addition to its role in the inflammatory status of adipose tissue during obesity, NOX2 regulates bone metabolism. Compared with obese wild-type mice, obese NOX2-knockout mice presented higher BMD and lower pro-osteoclastic factors, RANKL and cathepsin K. The bone marrow precursors derived from NOX2-knockout mice showed a reduced capacity to differentiate into osteoclasts in vitro, compared to wild-type cells [213]. Corroborating this data, Kang and Kim (2016) showed the role of NOX2 in promoting osteoclast differentiation through NFATC1 upregulation [214]. The data indicate that signaling mechanisms involving NOX2 activity mediate bone resorption during obesity.

Deficiency in antioxidant responses also contributes to oxidative stress and, consequently, unbalancing bone remodeling during obesity, which decreases glutathione peroxidases (GPX) [207,215,216]. HFD-induced obesity lowers the GPX3 plasma concentration in mice [207] and GPX1 expression in the liver [215] and reduces serum glutathione [216]. Hu and collaborators (2021) described that GPX7 favored osteogenic differentiation of MSCs and inhibited adipogenic commitment [217]. Obese GPX7 knockout mice under HFD presented increased inguinal, gonadal, and mesenteric fat accrual and higher plasmatic FFA concentration than wild-type obese mice [218]. An in silico analysis predicted that GPX7 expression is downregulated by miR 335-5p, which is upregulated in white adipose tissue of obese mice [219]. Thus, GPX7 may be a relevant target for oxidative stress-associated bone loss. However, GPX activities in obesity and their relationship with bone tissue regulation must be better elucidated.

Antioxidants may be a potential therapeutic approach to reduce the negative impacts of oxidative stress promoted by obesity on bones. The administration of N-acetylcysteine (NAC), a glutathione precursor, in HFD mice contributed to increased bone mass, compared to non-treated obese animals. This effect was explained due to the raised reduced glutathione disposal and decreased osteoclast formation, both in vivo and in vitro [216]. Beneficial effects were also observed in HFD mice fed with folic acid (FA), which reduced osteoporosis, malondialdehyde (MDA), and FFAs levels in bone homogenates while increasing Nrf2 expression [220]. Nrf2 upregulation in osteoblasts reduced IL-6 secretion, decreasing osteoclast differentiation in vitro [221]. In addition, green tea polyphenol supplementation in obese mice stimulated both BMD/bone strength and GTX1 expression in the liver and diminished fat mass and proinflammatory mediators [215]. Sodium butyrate, an antioxidant, ameliorated the metabolic profile of obese rats, enhancing mitochondrial antioxidant enzymes and redox homeostasis and improving bone condition [222]. Together, the data reinforce that oxidative stress promoted by obesity is a critical mechanism in bone loss, becoming a target of pharmacological strategies to alleviate osteoporosis.

### 3.7. Free and Vesiculated Noncoding RNAs

In addition to the protein-coding function that RNAs can assume, non-coding RNAs (Nc-RNA), often underestimated, have recently gained notoriety for their transcriptional and post-translational regulatory capacity. Nc-RNA is a large group of RNA sequences divided into two groups: long non-coding RNA (Lnc-RNA) and short non-coding RNA (Snc-RNA). Snc-RNA is formed by MicroRNA (MiRNA), small interfering RNA (siRNA), small nucleolar RNA (snoRNA), ribosomal RNA (rRNA), transfer RNA (tRNA), and piwi-interacting RNA (piRNA). Inc-RNA group comprises competing endogenous RNA (CeRNA), circular RNA (CircRNA), long intergenic non-coding RNA (lincRNA), antisense RNA, and pseudogenes [223,224].

The MiRNAs are one of the main subclasses of Snc-RNAs and control the expression of specific proteins through pairing with messenger RNAs (mRNA). Once paired with a complementary mRNA, MiRNA can induce mRNA cleavage or repress its translation on ribosomes, which blocks some protein synthesis [27]. CeRNAs, a subclass of Lnc-RNA, directly control the activity of MiRNAs and act like a “sponge,” competitively sequestering MiRNAs and inhibiting their interaction with mRNAs, reestablishing the expression of silenced genes [225].

In addition to their physiological functions, Nc-RNAs also contribute to the development of obesity, cancer, cardiovascular, autoimmune, and bone diseases. These Nc-RNAs released in their free form or carried by extracellular vesicles (EVs), are considered valuable for diagnostic strategies [226,227,228,229,230]. Herein, we have focused on five MiRNAs and one CeRNA that are described to be up-or downregulated in obesity and able to interfere in bone remodeling, becoming potential ways to induce osteoporosis. All selected data are comprised in Table 1.

MiR-34a-5p is upregulated in the plasma of subjects with obesity and late-state osteoarthritis, compared with healthy controls; furthermore, it is also upregulated in the plasma and knee joint of HFD mice [231]. It is known that HFD mice released increased amounts of derived exosomes enriched in MiR-34a, contributing to adipose tissue inflammation and inhibiting the M2 polarization of macrophages [232]. Yuan and collaborators (2021) demonstrated that MiR-34a-5p inhibits MSCs’ osteogenic differentiation [233]. A study analyzing old Brazilian adults demonstrated that high circulating MiR-34a-5p was associated with lower BMD in the lumbosacral region [234]. MiR-155 was implicated in osteogenesis inhibition through the regulation of SMAD5 [235], and SOCS1 [236] expression was detected in exosomes derived from obese adipose tissue and promoted insulin resistance [237]. Microvesicles derived from HFD mice’s adipocytes also contain more copies of MiR-155 [238]. In contrast, MiR-503, described as an enhancer of osteoblast differentiation [239] and inhibitor of adipogenesis [240], is downregulated in the plasma of subjects with obesity and is associated with a poor metabolic state [241]. An exciting work from Li and collaborators (2019) described MiR-149-3p as a regulator of MSC fate, inhibiting adipogenesis and favoring osteogenesis; its overexpression decreased obesity-associated gene (FTO) expression [242]. However, it is still unclear whether MiR-149-3p is overexpressed/suppressed or unchanged in patients with obesity.

Regarding the impact on osteoclast formation, two MiRNAs upregulated during obesity stand out, MiR-142-5p and MiR-155 [237,238,243]. MiR-142-5p targets PTEN-mediated signaling and increases bone marrow macrophage differentiation toward osteoclasts [244]. This MiRNA also reduces MSCs migration through VCAM1, contributing to osteoporosis [245]. MiRNA MiR-155, besides its effect on osteoblast-adipocyte regulation, can enhance osteoclastogenesis, targeting leptin receptor LepR expression [246]. In contrast, Zhang and colleagues (2012) showed an inhibitory effect of MiR-155 on osteoclast formation through modulation of IFN-β expression [247].

**Table 1 cells-12-00521-t001:** Obesity-associated non-coding RNAs and their impacts on bone remodeling. Obese adipose tissue releases non-coding RNAs that promote enhanced formation of osteoclasts and adipocytes while decreasing bone formation and osteoblast differentiation. These non-coding RNAs could be released freely or inside extracellular vesicles, derived mainly by adipocytes and macrophages from obese adipose tissue.

Noncoding RNA	Target	Effect on Bone Cells	Obesity Status	Free or Vesiculated
MiR-34a-5p	WNT1 [233]	Inhibition of Osteogenesis [233] Decreased Bone Mineral Density [234]	Upregulated [231,232]	Free [231] Obese Adipocyte-derived Exosomes [232]
MiR-142-5p	VCAM-1 [245] PTEN [244]	Inhibits Bone Marrow MSCs Migration Leading to Osteoporosis [245] Favors Osteoclast Differentiation [244]	Upregulated [243]	Free
MiR-155	SMAD5 [235] SOCS1 [236] LepR [246] IFN-β [247]	Inhibit Osteogenesis [235,236] Favors Osteoclast Differentiation [246] Inhibits Osteoclast Differentiation [247]	Upregulated [237,238]	Obese Adipose Tissue Macrophages-Derived Exosomes [237] Obese Adipocyte-Derived Microvesicles [238]
MiR-503	BMPR1a [240] SMURF1 [239]	Inhibit Adipogenesis [240] Favors Osteogenesis [239]	Downregulated [241]	Free
Lnc-H19	Mir-467/ HOX10 [248]	Favors Osteogenesis [248] Inhibits Adipogenesis [248]	Downregulated [248]	Mesenchymal Stem Cell-Derived Exosomes
MiR-149-3p	Obesity-Associated Gene FTO [242]	Favors Osteogenesis [242] Inhibit Adipogenesis [242]	Unknown Overexpression of FTO Gene is Associated with Obesity Development [249]	Unknown

A CeRNA, Inc-H19, contributed to obesity-associated bone loss, acting as a competitive inhibitor binding to complementary MiRNAs, and blocking its actions. It was shown that HFD mice presented lower amounts of MSC-derived exosomes, with less Inc-H19 content than normal diet mice. Exosomes containing Lnc-H19 promoted osteogenesis and repressed adipogenesis in vitro. This action was mediated through MiR-467 inhibition, which modulates HOX10 expression [248].

In the set, the data highlight that during obesity, four MiRNAs and one Lnc-RNA, released by adipose tissue, contribute as additional mechanisms of bone loss by repressing osteogenesis and favoring adipogenesis and osteoclast formation. Additional studies should be conducted to identify new classes of Nc-RNAs in subjects with obesity that can interfere with bone metabolism and may contribute to developing new bone loss biomarkers and therapies for obesity osteoporotic state.

## 4. Conclusions

Obesity is an inflammatory disease that may promote loss of quality of life. The release of several proinflammatory factors by the adipose tissue induces the onset of other pathological conditions, such as diabetes, atherosclerosis, osteoarthritis, and cancers. In bones, obesity unbalances osteoblast and osteoclast activities leading to bone loss. Among the mechanisms contributing to obesity-associated osteoporosis, the following stand out: proinflammatory cytokines (IL-1β, IL-6, and TNF-α) and adipokines (leptin and adiponectin); changes in gut microbiota and intestinal permeability; insulin resistance; oxidative stress; and free or EV-carried non-coding RNAs. The assessment of all of these mechanisms is relevant for understanding the harmful effects of obesity and developing new anti-obesity and anti-osteoporosis therapeutic strategies.

## Figures and Tables

**Figure 1 cells-12-00521-f001:**
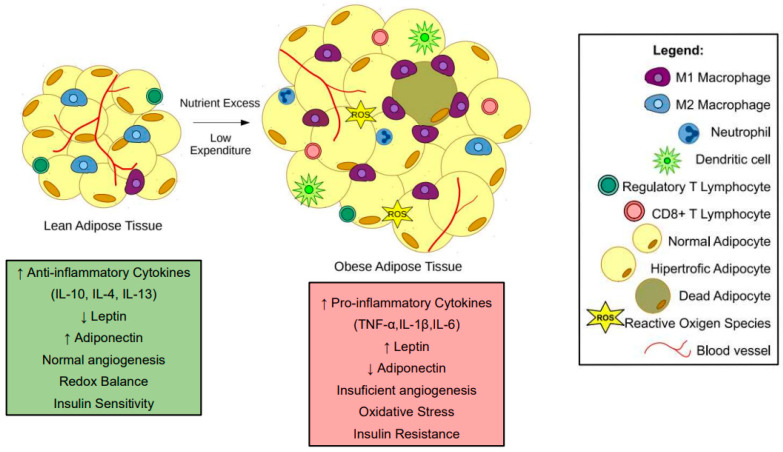
Changes in adipose tissue during obesity. In lean subjects, adipose tissue presents an anti-inflammatory state. Moreover, adipocytes present average triglyceride storage. During obesity, nutrient excess and low expenditure promote a hypertrophic state in adipocytes. The hypertrophic adipocytes and pro-inflammatory immune cells, such as M1-like macrophages, neutrophils, and CD8+ T cells, contribute to a low-grade inflammatory state in subjects with obesity. Most of the pro-inflammatory factors released by obese adipose tissue interfere with bone cells leading to bone loss.

**Figure 2 cells-12-00521-f002:**
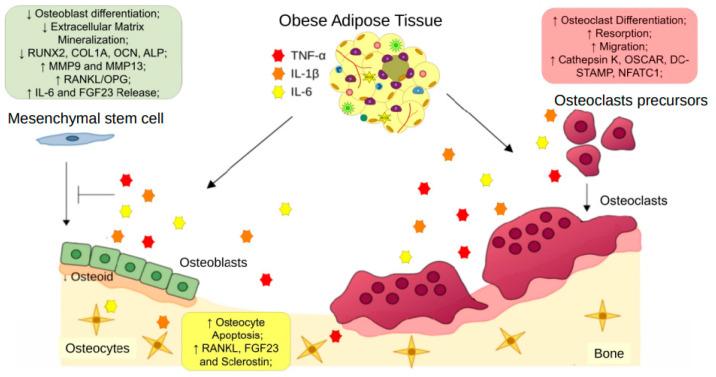
Obese adipose tissue-derived proinflammatory cytokines on bone remodeling. Pro-inflammatory cytokines, such as TNF-α, IL-6, and IL-1β, released by obese adipose tissue, exert effects on osteoclast precursors, increasing osteoclast differentiation and resorption. Moreover, these cytokines reduce the osteogenic differentiation of mesenchymal stem cells and compromise osteoblast bone formation. TNF-α, IL-6, and IL-1β also promote apoptosis and increased expression of osteoclast stimulation factors by osteocytes. The unbalanced activities between osteoblasts and osteoclasts induce osteoporosis.

**Figure 3 cells-12-00521-f003:**
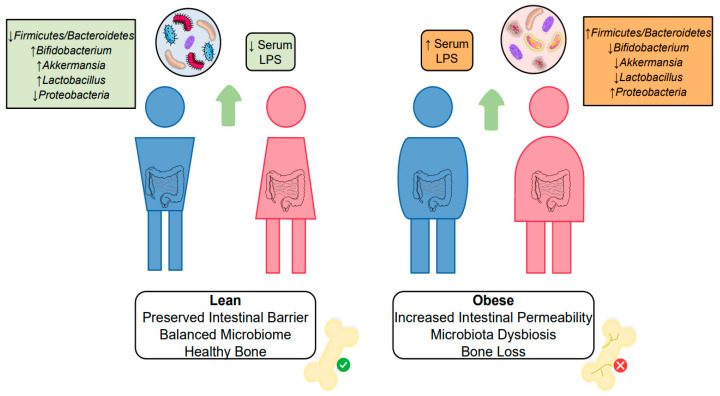
Impacts of gut alterations and dysbiosis on bone remodeling. Obesity promotes an increased permeability of the intestinal barrier, which leads to enhanced serum LPS concentrations. High endotoxin levels and the altered gut microbiota, with increased Firmicutes/Bacteroidetes ratio and decreased frequency of beneficial microorganisms, contribute to increased bone resorption and decreased bone formation in obesity.

**Figure 4 cells-12-00521-f004:**
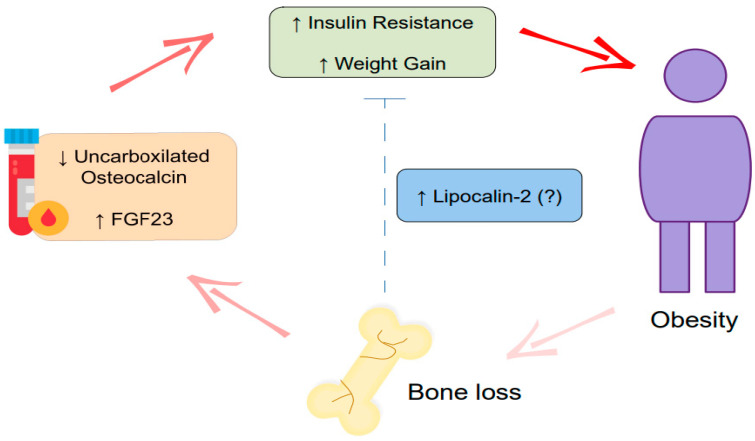
Obesity’s effects on osteokines secretion and bone loss. Obesity promotes bone dysfunctions that alter osteokine secretion. Serum uncarboxylated osteocalcin is reduced in subjects with obesity, while FGF-23 concentration is enhanced. Thus, this pathologic condition aggravates insulin resistance and weight gain. The worsening of obesity leads to more severe bone damage, becoming a vicious cycle. Lipocalin-2, which is also produced by bone cells, may exert anti-obesity effects, but the participation of lipocalin 2 in obesity-associated bone loss still needs to be fully understood.

## Data Availability

Not applicable.

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
