# Peer review of "Cellular and Molecular Mechanisms Associating Obesity to Bone Loss"

_cells, 2023, doi:10.3390/cells12040521_

Round 1

Reviewer 1 Report

The authors have carried out a detailed analysis of the molecular mechanisms associated with bone loss in obesity.

This is a very well-documented review that provides the latest studies on the subject. The figures are very illustrative.

However, I have some suggestions:

-First of all, throughout the article "obese patient" is written, which is currently not acceptable instead please change to “person or patients with obesity”.

-Regarding the relationships of hormones with bone, the paper only refers to leptin/adiponectin. It would be worth talking about ghrelin and GLP-1, even if only briefly, especially nowadays with the rise of treatments with GLP-1 receptor analogues.

-On the other hand, a section should be added on the effects of sarcopenia and intramuscular adipose tissue and its negative influence on bone due to inflammation/myokine secretion.

Author Response

REVIEWER #1

R: The authors have carried out a detailed analysis of the molecular mechanisms associated with bone loss in obesity. This is a very well-documented review that provides the latest studies on the subject. The figures are very illustrative. However, I have some suggestions:

 A: We thank the reviewer for her/his suggestions, which have improved the manuscript. All points were fully addressed, as commented below.

R: First of all, throughout the article "obese patient" is written, which is currently not acceptable instead please change to “person or patients with obesity”.

A: We apologize for that mistake. The term "obese patient" was substituted as suggested.

 R: Regarding the relationships of hormones with bone, the paper only refers to leptin/adiponectin. It would be worth talking about ghrelin and GLP-1, even if only briefly, especially nowadays with the rise of treatments with GLP-1 receptor analogues.

 A: Thank you for your comment. As recommended, we have provided a complementary text on ghrelin and GLP-1 receptor analogs, which was included in the manuscript- in topic 3.1. 

  -On the other hand, a section should be added on the effects of sarcopenia and intramuscular adipose tissue and its negative influence on bone due to inflammation/myokine secretion.

A: Thank you for your suggestion. We have provided a new topic (3.3) entitled Muscle-fat-bone crosstalk during obesity to address this point.

Following the suggestions, we have also included new references in the manuscript.

Text Included in the manuscript: (PAGES 7-8)

3.2. Adipokines in bone remodeling:

…….

In addition to leptin and adiponectin, Ghrelin and Glucagon-like peptide 1 (GLP-1), are altered during obesity also influence bone metabolism [122], and may have a role in bone remodeling. Ghrelin is produced by the stomach, acting on appetite regulation and energy metabolism, improves cardiovascular function, and can act as on bone formation [123].  Erener et al. (2021) showed that ghrelin administration in mice enhanced the healing process of bone fractures [123]. Notably, in patients with osteopenia the systemic ghrelin concentration are inversely related to BMI values, and the osteoblasts isolated from those patients were resistant to ghrelin-mediated signaling [124].

The therapeutic use of GLP-1 receptor agonists has been well described to improve for the metabolic parametrers of patients with obesity and insulin resistance [125]. Additionally, activation of GLP-1 receptor on MSCs induced osteogenic differentiation via PKA/β-catenin signaling activation [126], and improved bone composition in obese animals [127]. Together, the data point to ghrelin and GLP-1 as worthy candidates for pharmacological intervention for controlling obesity-associated bone loss.

3.3 Muscle-fat-bone crosstalk during obesity

Bones and skeletal muscles are associated tissues that promote locomotion, and the mechanical forces driven by muscle contraction are vital factors that allow bone formation. However, the muscle-bone crosstalk is not limited to structural functions. Miokines produced and released in response to muscular contractions have osteoanabolic effects and are essential for maintaining bone metabolism [128]. Chronic low-grade inflammation in patients with obesity modifies the crosstalk between muscle-fat-bone [129]. Fatty acids released by intramuscular adipose tissue induced lipotoxic effects on osteoblasts, osteocytes, and myocytes [130].  Disturbances in muscle metabolism caused by overweight/obesity may impact bone tissue, contributing to osteosarcopenic obesity syndrome.

 The sedentary profile of people with obesity reduces the anti-inflammatory activity performed by myokines, such as Irisin. This exercise-triggered myokine promotes white adipose tissue browning, energy expenditure and stimulates bone formation [131]. Knockout mice for the Irisin precursor FNDC5 (Fibronectin‐type III domain‐containing 5) showed bone loss mediated by increased resorption activity and reduced bone formation [132, 133]. Irisin stimulates osteoblast differentiation through αV integrin activating-BMP/SMAD pathway [132], leading to activation of Runx1/2 via focal adhesion kinase [133]. The administration of Irisin in high-fat diet-induced obese mice increased insulin, ghrelin, and BMP4 circulating concentrations [134]. Obese rodents subjected to swimming showed improvement in bone loss, simultaneously with increased systemic irisin production and release [135]. Addressing the crosstalk among bone, muscle, and adipose tissue may be a successful pharmacological strategy for treating bone and muscle loss in obesity.

Reviewer 2 Report

The authors described the mechanisms of obesity-associated bone loss in detail, which are informative and good summarized to this study field. It would be better if can provide more discussion in every small part of Results or Conclusion, not just describe the published data.

Minor corrections needed:

1) line 64 "putative tools for diagnostic and therapeutic" need to add "s" "......diagnostics and therapeutics"

2) line 73 "reabsorb bone" should be "resorb bone"

3) line 323 " 3.2. Gut and Microbiota alterations" should be "3.3....."

Author Response

REVIEWER #2

R: The authors described the mechanisms of obesity-associated bone loss in detail, which are informative and well summarized in this study field. It would be better if can provide more discussion in every small part of the Results or Conclusion, not just describing the published data.

Minor corrections needed:

 1) line 64 "putative tools for diagnostic and therapeutic" need to add "s" "......diagnostics and therapeutics"

2) line 73 "reabsorb bone" should be "resorb bone"

3) line 323 " 3.2. Gut and Microbiota alterations" should be "3.3....."

 A: We thank the reviewer for his/her careful revision. All issues raised were addressed in the manuscript.

Text Included in the manuscript: (PAGES 7-8) (Suggestions Reviewer #1)

3.2. Adipokines in bone remodeling:

…….

In addition to leptin and adiponectin, Ghrelin and Glucagon-like peptide 1 (GLP-1), are altered during obesity also influence bone metabolism [122], and may have a role in bone remodeling. Ghrelin is produced by the stomach, acting on appetite regulation and energy metabolism, improves cardiovascular function, and can act as on bone formation [123].  Erener et al. (2021) showed that ghrelin administration in mice enhanced the healing process of bone fractures [123]. Notably, in patients with osteopenia the systemic ghrelin concentration are inversely related to BMI values, and the osteoblasts isolated from those patients were resistant to ghrelin-mediated signaling [124].

The therapeutic use of GLP-1 receptor agonists has been well described to improve for the metabolic parametrers of patients with obesity and insulin resistance [125]. Additionally, activation of GLP-1 receptor on MSCs induced osteogenic differentiation via PKA/β-catenin signaling activation [126], and improved bone composition in obese animals [127]. Together, the data point to ghrelin and GLP-1 as worthy candidates for pharmacological intervention for controlling obesity-associated bone loss.

3.3 Muscle-fat-bone crosstalk during obesity

Bones and skeletal muscles are associated tissues that promote locomotion, and the mechanical forces driven by muscle contraction are vital factors that allow bone formation. However, the muscle-bone crosstalk is not limited to structural functions. Miokines produced and released in response to muscular contractions have osteoanabolic effects and are essential for maintaining bone metabolism [128]. Chronic low-grade inflammation in patients with obesity modifies the crosstalk between muscle-fat-bone [129]. Fatty acids released by intramuscular adipose tissue induced lipotoxic effects on osteoblasts, osteocytes, and myocytes [130].  Disturbances in muscle metabolism caused by overweight/obesity may impact bone tissue, contributing to osteosarcopenic obesity syndrome.

 The sedentary profile of people with obesity reduces the anti-inflammatory activity performed by myokines, such as Irisin. This exercise-triggered myokine promotes white adipose tissue browning, energy expenditure and stimulates bone formation [131]. Knockout mice for the Irisin precursor FNDC5 (Fibronectin‐type III domain‐containing 5) showed bone loss mediated by increased resorption activity and reduced bone formation [132, 133]. Irisin stimulates osteoblast differentiation through αV integrin activating-BMP/SMAD pathway [132], leading to activation of Runx1/2 via focal adhesion kinase [133]. The administration of Irisin in high-fat diet-induced obese mice increased insulin, ghrelin, and BMP4 circulating concentrations [134]. Obese rodents subjected to swimming showed improvement in bone loss, simultaneously with increased systemic irisin production and release [135]. Addressing the crosstalk among bone, muscle, and adipose tissue may be a successful pharmacological strategy for treating bone and muscle loss in obesity.

3.4. Gut and Microbiota alterations influencing bone metabolism:……..